# Long-Term, Sex-Specific Effects of GCRsim and Gamma Irradiation on the Brains, Hearts, and Kidneys of Mice with Alzheimer’s Disease Mutations

**DOI:** 10.3390/ijms25168948

**Published:** 2024-08-16

**Authors:** Curran Varma, Maren K. Schroeder, Brittani R. Price, Khyrul A. Khan, Ernesto Curty da Costa, Camila Hochman-Mendez, Barbara J. Caldarone, Cynthia A. Lemere

**Affiliations:** 1Department of Neurology, Ann Romney Center for Neurologic Diseases, Brigham and Women’s Hospital, Boston, MA 02115, USA; cvarma@bwh.harvard.edu (C.V.); schroemk@odu.edu (M.K.S.); brittani.price@gehealthcare.com (B.R.P.); khankhyrul1996@gmail.com (K.A.K.); 2Department of Neurology, Harvard Medical School, Boston, MA 02115, USA; 3Department of Regenerative Medicine Research, Texas Heart Institute, Houston, TX 77030, USA; ecurty@texasheart.org (E.C.d.C.); cmendez@texasheart.org (C.H.-M.); 4Mouse Behavioral Core, Harvard Medical School, Boston, MA 02115, USA; barbara_caldarone@hms.harvard.edu

**Keywords:** GCRsim, gamma, radiation, Alzheimer’s disease, CNS, neurodegeneration, space radiation

## Abstract

Understanding the hazards of space radiation is imperative as astronauts begin voyaging on missions with increasing distances from Earth’s protective shield. Previous studies investigating the acute or long-term effects of specific ions comprising space radiation have revealed threats to organs generally considered radioresistant, like the brain, and have shown males to be more vulnerable than their female counterparts. However, astronauts will be exposed to a combination of ions that may result in additive effects differing from those of any one particle species. To better understand this nuance, we irradiated 4-month-old male and female, wild-type and Alzheimer’s-like mice with 0, 0.5, or 0.75 Gy galactic cosmic ray simulation (GCRsim) or 0, 0.75, or 2 Gy gamma radiation (wild-type only). At 11 months, mice underwent brain and heart MRIs or behavioral tests, after which they were euthanized to assess amyloid-beta pathology, heart and kidney gene expression and fibrosis, and plasma cytokines. Although there were no changes in amyloid-beta pathology, we observed many differences in brain MRIs and behavior, including opposite effects of GCRsim on motor coordination in male and female transgenic mice. Additionally, several genes demonstrated persistent changes in the heart and kidney. Overall, we found sex- and genotype-specific, long-term effects of GCRsim and gamma radiation on the brain, heart, and kidney.

## 1. Introduction

NASA’s Artemis program has set the stage for humankind’s return to venturing beyond low Earth orbit and the protection of Earth’s magnetic shield. With Artemis II expected to set the record for the farthest human travel from Earth, followed by Artemis III’s mission to return to the lunar surface, it is essential we understand how the environmental risks of space affect human biology as we set our sights on even farther targets, like Mars. Chronic exposure to high-energy ionizing radiation, such as gamma rays, originating from solar particle events from the sun and omnidirectional galactic cosmic rays (GCRs), presents one of the largest hazards to astronaut health. Chest X-rays are an example of a type of ionizing radiation humans are frequently exposed to and are similar to gamma rays; however, astronauts are subjected to radiation doses ranging from the equivalent of 150 to 6000 chest X-rays [1], in addition to many other types of radiation. Furthermore, space radiation also consists of a cocktail of high-charge and -energy ions that can cause secondary radiation upon impact with a surface. The question then becomes: what happens when these ionized particles, moving at nearly the speed of light, pass through different tissues?

The brain and heart are both organs consisting of cells that do not readily divide, such as neurons and cardiomyocytes, respectively, and are believed to be relatively radioresistant. However, the chronic exposure astronauts endure has warranted a closer look in recent years. To better understand whether astronauts have an increased risk of later developing Alzheimer’s disease (AD) upon return to Earth, our lab, alongside others, has previously investigated the effects of specific particles, such as protons [2] and ^56^Fe [3,4,5], which are major contributors to space radiation, on the central nervous system (CNS) in rodents. While previous work has revealed the cellular effects of ^56^Fe on double-strand breaks and apoptosis in human monocytes [6], interestingly, the results of our studies have indeed demonstrated alterations in neurobehavioral tests, inflammatory markers, and brain pathology with irradiation, suggesting that radiation has harmful impacts on the CNS and the potential to promote neurodegeneration.

Recent technological advancements have made it possible to simulate GCR (GCRsim) using a spectrum of ion beams delivering specified proportions of radiation sequentially [7,8], allowing us to utilize 5-ion GCRsim: a combination of protons, helium, silicon, oxygen, and iron. This innovation has allowed us to more closely replicate the deep space environments astronauts will journey through on their way to Mars. Several studies have taken advantage of this method to study changes in the brain [9,10,11,12,13,14,15,16], heart [17], kidneys [18], and lungs [19], and though modest effects were seen in the heart, various neurobehavioral and cognitive tests have revealed sex-specific differences in response to GCRsim where male rodents are disproportionally affected and demonstrate greater cognitive deficits compared to females, who may be protected by estrogen [9]. Given that biological sex is highly relevant to human physiology under both normal and pathological conditions, such as AD, and that one of the goals of the Artemis program is to land the first woman on the Moon, it is imperative that we understand the sex-specific risks of space radiation.

Though previous studies have helped lay the groundwork in showing GCRsim to have sex-specific neurological effects on rodents, none thus far have investigated its effects in an AD mouse model. To explore the long-term effects of GCRsim on AD risk in a mouse model with AD mutations that mimic amyloid pathology and gliosis and develop cognitive impairment by 7 months of age [20], and determine its sex-specificity, we compared the effects of whole-body exposure to 0, 0.50, or 0.75 Gy GCRsim on 4-month-old male and female wild-type (WT) and APPswe/PS1dE9 transgenic mice (Tg) as well as gamma irradiation (0.75 and 2 Gy) on WT mice for extrapolation to humans. In addition to conducting a battery of behavioral tests, we also investigated differences in amyloid-beta (Aβ) pathology, a hallmark of AD, pro-inflammatory cytokines in plasma, volumes of several brain regions via MRI neuroimaging, cardiac structure and function via dynamic MRI imaging, and gene expression in hearts and kidneys. Behavioral tests, as well as MRI analyses of the brain and heart, all began around 11 months of age, approximately 7 months after irradiation, and allowed us to assess the late CNS and cardiovascular consequences of GCRsim.

## 2. Results

### 2.1. GCRsim and Gamma Irradiation Do Not Affect Survival Rate

To investigate the late effects of whole-body GCRsim and gamma irradiation, male (M) and female (F) WT and Tg mice were sent to the NASA Space Radiation Laboratory at Brookhaven National Laboratory at 4 months of age and were comprehensively evaluated 7 months later (Figure 1A). 

When comparing survival rates, no radiation-specific effects were seen. However, sham-irradiated Tg females demonstrated a significantly lower survival rate compared to both their wild-type (Figure 1B, *p* = 0.0041) and male counterparts (Figure 1B, *p* = 0.0437). Additionally, highly significant increases in body weight at 12 months of age were observed compared to 4 months in all groups; however, this is likely due to the aging process, as increases in body weights were consistent across all groups (Figure 1D,E).

### 2.2. Cognitive and Anxiety-like Behaviors

Overall, WT mice appeared to be more susceptible to radiation-induced cognitive changes compared to their Tg counterparts. For example, in the spatial novelty Y maze, M WT mice irradiated with 0.75 Gy GCRsim spent a lower percentage of time in the novel arm compared to sham-irradiated controls, indicating impaired spatial memory (Figure 2C, *p* = 0.0057); however, this effect was not seen in M Tg mice. Similarly, M WT mice that received 2 Gy gamma irradiation also spent less time (Figure 2D, *p* = 0.0205) and traveled less distance (Figure 2B, *p* = 0.0653) in the novel arm compared to the sham-irradiated group, suggesting that gamma exposure induced some cognitive impairment in M WT mice. No radiation-specific effects were seen in fear memory in the contextual fear conditioning test with either GCRsim or gamma irradiation (Figure 2E,F). When assessing depressive-like behavior through the tail suspension test, no radiation effects were found in relation to GCRsim or gamma. However, there was a significant genotype effect where F Tg sham-irradiated mice spent less time immobile compared to WT (Figure 2G, *p* = 0.0004), indicating less stress-induced behavior. The elevated plus maze revealed a dose-dependent relationship with M Tg mice, where the 0.5 Gy GCRsim group spent relatively more time in the open arm compared to sham-irradiated mice and significantly more than the 0.75 Gy mice (Figure 2I, *p* = 0.0326), which suggests that the mice receiving 0.5 Gy GCRsim experienced reduced anxiety-like behavior. This relationship was not seen in the M WT gamma groups, where the percent time in the open arm was more consistent across doses (Figure 2J). On the other hand, the percent distance traveled in the center of the open field, another test of anxiety-like behavior was not significantly different across doses for M Tg mice receiving GCRsim (Figure 2M), but we did find a trend for a reduction in M WT mice receiving 0.75 Gy gamma (Figure 2N, *p* = 0.0939), pointing to increased anxiety-like behavior. Additionally, there was a significant sex difference in sham-irradiated WT mice with F WT mice naturally demonstrating greater anxiety-like behavior than M WT (Figure 2M,N, *p* = 0.0145). 

Both GCRsim and gamma were able to lower pre-pulse inhibition at 82 decibels in F WT mice, which indicates an impaired ability to filter sensory stimuli. Reduced inhibition trended towards significance with F WT mice receiving 0.75 Gy GCRsim (Figure 2O, *p* = 0.0872) but reached significance with 2 Gy gamma (Figure 2P, *p* = 0.0119) when compared to shams. Overall, both GCRsim and gamma irradiation impaired sensorimotor gating in F WT mice, and we found a trend for a sex difference in terms of reduced inhibition in M WT shams (Figure 2O,P, *p* = 0.0594). With the startle test, different white noise tones revealed radiation-specific effects on startle response. For example, while there was a trend for reduced startle response at 90 dB in WT males receiving 0.5 Gy GCRsim compared to sham controls (Figure 2Q, *p* = 0.0858), Tg males in the 0.75 Gy GCRsim group demonstrated significantly lower reactivity than those in the 0.5 Gy group (Figure 2Q, *p* = 0.0465) at 60 dB. Furthermore, gamma irradiation (2 Gy) also reduced startle response at 70 dB in WT F mice (Figure 2R, *p* = 0.0143). Our findings from the startle test suggest that radiation can impact dopaminergic pathways with higher doses, increasing the likelihood of this occurrence. Lastly, there were no significant differences found in terms of fear memory (Figure 2E,F) in relation to irradiation.

### 2.3. Non-Cognitive Behaviors

We found both GCRsim and gamma irradiation to have many effects on non-cognitive behaviors, such as locomotor activity, endurance, and coordination. For example, several GCRsim-induced effects were found, with the average latency to fall in the rotarod test. F Tg mice receiving 0.75 Gy GCRsim displayed a trend towards lower latency to fall compared to the sham-irradiated group (Figure 3I, *p* = 0.0804) but had a significantly lower latency compared to the 0.5 Gy group (Figure 3I, *p* = 0.0364). Additionally, in M Tg mice, there was a trend towards an elevated latency to fall in the 0.75 Gy GCRsim group compared to both the sham-irradiated (Figure 3I, *p* = 0.0639) and 0.5 Gy group (Figure 3I, *p* = 0.0645). Interestingly, it appeared that 0.75 Gy GCRsim impaired motor coordination in F Tg mice but improved it in M Tg mice, pointing to a sex difference in how radiation affects the motor system. Furthermore, these effects may be dependent on the type of radiation as they were not observed with gamma irradiation (Figure 3J). Additional sex- and genotype-specific effects were found as F Tg mice demonstrated a greater latency to fall compared to both F WT (Figure 3I, *p* = 0.0198) and M Tg (Figure 3I, *p* = 0.0017) mice; however, this is not surprising as F APP/PS1 mice are known to be hyperactive [2,4,5] which we speculate may have influenced a quicker release. When looking at the percent improvement on the rotarod test, we again noted a difference between F Tg mice in the 0.5 and 0.75 Gy GCRsim groups, where 0.75 Gy mice displayed a near significant improvement (Figure 3K, *p* = 0.0668), potentially indicating that the higher dose of radiation impairs motor coordination but boosts motor learning. With this assessment, there were also GCRsim-induced changes in the F WT mice, as both the 0.5 and 0.75 Gy groups trended towards a reduced percent improvement (Figure 3K, *p* = 0.0743, *p* = 0.0751, respectively). Again, there were no effects seen with gamma radiation (Figure 3L). 

### 2.4. Beta-Amyloid Pathology

Somewhat surprisingly, unlike in our previous studies, there were no radiation-specific effects on Aβ as assessed through insoluble ELISAs and immunostaining in Tg mice (Figure 4A–F). Trends for sex differences in insoluble Aβ42 levels (Figure 4B, *p* = 0.0979) and Thio S staining (Figure 4C,D, *p* = 0.0009) were found with sham-irradiated Tg mice, with females having higher levels for both, but no overall radiation-specific effects were seen. 

### 2.5. Brain MRI Volumetric Analyses

Although there were no changes in AD amyloid pathology due to radiation, after follow-up MRI scans, we did find several radiation-induced changes in the volumes of the lateral ventricles, hippocampus, and cortex of the brain with both GCRsim and gamma (Appendix A). Neurodegeneration causes enlargement of the lateral ventricles, and through analyses of T1 and T2 brain MRI scans, we found increases in the volumes of the lateral ventricles in several groups with GCRsim. For example, while aging did not cause a significant increase in ventricular volumes in sham-irradiated F WT mice, 0.5 Gy GCRsim caused significant enlargement of the ventricles several months after irradiation (Figure 5A, *p* = 0.0173). Interestingly, this radiation effect was not found with 0.75 Gy GCRsim, F Tg mice, or gamma irradiation. GCRsim also impacted M Tg mice, as both the 0.5 Gy (Figure 5A, *p* = 0.0476) and 0.75 Gy (Figure 5A, *p* = 0.0149) radiation groups experienced a significant elevation in ventricular volumes; however, no effects were seen in the F Tg groups. The sham-irradiated M WT mice had the greatest difference between the baseline and follow-up scans (Figure 5A, *p* = 0.0014), though this is likely due to aging. Gamma irradiation did not impact the lateral ventricles of F or M WT mice, but given that sham-irradiated WT mice were shared between the GCRsim and gamma analyses, the sham-irradiated M WT group was again significant (Figure 5B, *p* = 0.0254). 

When investigating volumetric differences in the hippocampus, which plays a role in memory, learning, and emotion, we observed that males were differentially impacted by GCRsim. While both F WT and Tg mice were unaffected by GCRsim, their male counterparts were, for example, 0.75 Gy GCRsim increased the hippocampal volume of M WT mice (Figure 5C, *p* = 0.0352), which was not seen in the sham controls. As seen with the lateral ventricles, GCRsim again induced changes in the brains of 0.5 Gy (Figure 5C, *p* = 0.0132) and 0.75 Gy (Figure 5C, *p* = 0.0118) M Tg mice by elevating hippocampal volumes, although these effects are contrary to those in the ventricles. Furthermore, similar changes were seen in both male and female WT mice receiving 0.75 Gy and 2 Gy gamma, respectively (Figure 5D, *p* = 0.034, *p* = 0.0417). Given that these effects were not seen in the sham controls, these results demonstrate gamma radiation’s ability to affect the hippocampi of both sexes, as opposed to GCRsim, which only impacted that of males. However, this was not the case with the cortex, as exposure to 0.75 Gy GCRsim reduced the cortical volumes of both male and female WT mice (Figure 5E, *p* = 0.0009, *p* = 0.0099, respectively). However, there was no aging effect in F WT mice, making this finding more clearly attributable to radiation. Sham-irradiated M WT mice trended towards reduced cortical volumes (Figure 5E, *p* = 0.098), and it appears that GCRsim has the potential to bolster this effect at higher doses. Gamma irradiation was especially impactful on the cortex, though there seemed to be an interesting relationship between dose and sex. In WT females, cortical reductions increased with higher doses of gamma radiation (Figure 5F, *p* = 0.0346, *p* = 0.002, respectively); however, in males, the opposite appears true, as differences in baseline and post-irradiation volumes became smaller with increasing doses (Figure 5F, *p* = 0.0073, *p* = 0.0453, respectively) and suggest a protective effect. These results exemplify sex-specific changes in relation to radiation. Overall, we found inconsistent radiation effects across several brain regions. For example, with M Tg mice, both doses of GCRsim enlarged the lateral ventricles; however, given that there were no reductions in cortical volumes, it is difficult to conclude that GCRsim induced neurodegeneration and brain atrophy, especially when hippocampal volumes increased. 

### 2.6. Cardiac MRI Structural and Functional Analyses

Next, we examined the effects of GCRsim and gamma irradiation on cardiac structure and function. To investigate this, we utilized short-axis FLASH MRI imaging to measure changes in left ventricular mass index, ejection fraction, stroke volume index, and ventricular wall thickness (Appendix A). As opposed to the brain, few radiation-specific effects were seen with the heart, where there were no significant changes in ejection fraction, stroke volume index, or ventricular wall index due to GCRsim or gamma exposure (Figure 6C–H). However, we did observe a trend for reduced left ventricular mass index in F WT mice receiving 0.75 Gy gamma (Figure 6B, *p* = 0.061), pointing to cardiac muscle weakening. Sham-irradiated M WT mice, shared between the GCRsim and gamma analyses, also experienced a significant reduction (Figure 6A,B, *p* = 0.0076, *p* = 0.012, respectively), which may have been due to the aging process. Additionally, these changes were not seen with GCRsim or gamma irradiation, potentially indicating a protective effect.

### 2.7. Alterations in Heart and Kidney Gene Expression

To assess the enduring impact of GCRsim or gamma radiation exposure on gene expression in the heart and kidney, we conducted real-time PCR analysis targeting genes associated with fibrosis, energy metabolism, apoptosis, and inflammation. This comprehensive panel encompassed *CTGF*, *Col1A1*, *TGFB1*, *BMP3*, *IL6*, *VCAM1*, *OSTP* (heart-specific), *podocin* (kidney-specific), *ICAM1*, *Casp3*, *GLUT4*, *MCAD*, *VLCAD*, *IL17*, and *Trp53* (Figure 7A). This panel was selected based on gene functions. *VLCAD*, *MCAD*, and *GLUT4* were selected because of their role in energy metabolism. *ICAM*, *VCAM*, *IL-17*, and *IL-6* were selected because of their role in inflammation, especially vascular damage, which is hypothesized to play a role in the response to GCRsim. *Trp53* was selected because of its importance in the control of apoptosis. Evaluation of Tg mice with GCRsim revealed 46 statistically significant differentially expressed genes, with 24 in the heart and 22 in the kidney (Appendix A). However, upon considering the confidence intervals of fold change values, only three genes exhibited biologically meaningful differential expression: *VLCAD*, *GLUT4*, and *Casp3* (Figure 7B,C). Specifically, we noted a 0.65-fold decrease in *VLCAD* expression (Figure 7B, *p* = 0.0025, 95% CI 0.47–0.90) in the APP/PS1 genotype irradiated with the lower dose (0.5 Gy) of GCRsim, and a decrease of 0.57-fold (Figure 7B, *p* = 0.0014, 95% CI 0.37–0.87) in the 0.75 Gy group, when compared to sham-irradiated controls. Similarly, the WT genotype showed a parallel decrease, with a 0.49-fold reduction in *VLCAD* expression in the 0.5 Gy GCRsim group (Figure 7B, *p* = 0.0005, 95% CI 0.27–0.88). A similar decrease in *VLCAD* expression (0.44-fold) was also observed in the 0.75 Gy group (Figure 7B, *p* = 0.0004, 95% CI 0.20–0.98). Notably, the Tg mice exposed to the high-dose (0.75 Gy) of GCRsim exhibited downregulation of *Casp3* (Figure 7C, 0.27-fold, 95% CI 0.11–0.66, *p* = 0.0002) and *GLUT4* (Figure 7C, 0.41-fold, 95% CI 0.19–0.90, *p* = 0.001) relative to sham-treated mice. 

For the gamma-irradiated group, the only biologically meaningful result was a 0.25-fold reduction in the cardiac expression of *COL1A1* for the WT animals (*p* < 0.001, 95% CI 0.06–0.94; Appendix A, row #24). Despite the 8-month interval between GCRsim exposure and gene expression analysis, detectable differences in gene expression regulation persisted, suggesting potential long-term effects of GCRsim exposure on gene regulation. Given our finding with COL1A1, a marker of fibrosis [21], we were curious to stain with Picrosirius Red for fibrosis (Appendix A), which yielded minimal differences overall. However, an intriguing finding emerged, revealing a modest yet statistically significant reduction in the percentage of cardiac fibrotic area in APP/PS1 shams (0.7%) compared to WT shams (1.4%) (Appendix A, *p* = 0.03). Notably, within the WT genotype alone, there was an increase in the fibrotic area of the kidneys from 1.33% in the sham group to 2.1% in the 0.5 Gy GCRsim group (*p* = 0.042, Appendix A). No other statistically significant differences in terms of fibrosis were detected.

### 2.8. Plasma Cytokines

To better understand how GCRsim or gamma-induced changes in peripheral inflammatory cytokines, we performed ELISAs to measure cytokines in plasma. With the exceptions of IFN-γ (Figure 8A,B) and TNFα (Figure 8M,N), we observed many radiation-specific effects on inflammatory markers. For example, 0.75 Gy GCRsim induced a significant elevation in IL-2 in M Tg mice compared to the sham-irradiated group (Figure 8C, *p* = 0.0278), this relationship was not present in the female counterparts. This elevation appeared to be dependent on radiation type, as it was seen with GCRsim but not gamma irradiation (Figure 8D). Furthermore, 0.5 Gy GCRsim induced a reduction in both IL-5 (Figure 8E, *p* = 0.0167) and IL-6 (Figure 8G, *p* = 0.033) in M Tg mice that, again, did not occur with gamma irradiation. These findings may potentially be due to a combined effect of genotype and radiation. Additionally, levels of IL-5 were naturally higher in F WT mice compared to males (Figure 8E, *p* = 0.0086). 

With M WT mice, 0.75 Gy GCRsim caused a near-significant reduction in levels of IL-10 compared to 0.5 Gy and sham-irradiated mice (Figure 8I, *p* = 0.0976, *p* = 0.0508, respectively), whereas levels were more consistent across doses of gamma (Figure 8J). However, levels of KC/GRO in 2 Gy gamma-treated F WT mice did trend towards a significant elevation compared to the sham group (Figure 8L, *p* = 0.0939), the only radiation-specific effect on cytokines in females. Additionally, KC/GRO was the only cytokine to be impacted solely by gamma irradiation (Figure 8K). Overall, the radiation-induced changes we observed in plasma cytokines were almost exclusively in males, indicating their increased susceptibility to radiation-induced inflammation and supporting the implications of biological sex.

## 3. Discussion

Space radiation is a major contributor to the uniquely hazardous environment of space, and its biological effects can be significant and concerning. Our previous studies, along with others, investigating the isolated effects of single-ion radiation, such as ^56^Fe or protons, have provided valuable insights in terms of understanding the potential impact that specific particles may have. However, in space, astronauts endure chronic radiation consisting of a combination of ions, including the ions represented in the five-ion GCRsim, which can be dramatically more complex due to additive effects between the different particles. To better understand the potential risks astronauts will face on their journey through deep space to destinations like Mars, and perhaps one day, beyond, we sought to investigate changes in behavior, biochemistry, and pathology using doses of space radiation relevant to a round trip to Mars [7]. This study is one of the first to assess the long-term effects of GCRsim and gamma irradiation on the CNS and cardiovascular system using APP/PS1 Tg and WT male and female mice. Additionally, a key component of this study was analyzing the role of sex on our endpoints, which we have learned can inform differential risks for female and male astronauts. 

Overall, our behavior results from both GCRsim and gamma irradiation further supported evidence of sex differences in relation to radiation, as males were disproportionally affected across our battery of tests. For example, with the spatial novelty Y-maze, we found that both GCRsim and gamma irradiation impaired spatial memory in M WT mice, as seen through reduced percent time spent in the novel arm. However, regardless of genotype and radiation dose/type, females displayed no differences. These findings align with a previous study reporting that male mice exposed to 50 cGy (0.5 Gy) GCRsim had impairments in spatial learning, which were not seen in females [10]. Additionally, the study found that females exposed to the same dose of radiation as males made fewer errors in the radial arm water maze, further demonstrating the sex-dimorphic response. These results may be explained by another study showing that both acute and chronic GCRsim exposure diminished hippocampal synaptic plasticity, essential for learning and memory [22].

We observed few effects on anxiolytic and depressive-like behavior. On the elevated plus maze, 0.75 Gy GCRsim M Tg mice spent a lower percentage of time in the open arm compared to the 0.5 Gy group; however, there was no significance compared to the control group. While an n-shaped dose–response curve may be at play, it is also possible that these effects are time-dependent, as another study implementing 3-ion GCRsim found that male mice demonstrated increased anxiety 80 days post-irradiation but not at the 45-day mark [9]. Still, other studies investigating ^56^Fe or proton irradiation [2,5,23,24] also found minimal effects of radiation on these outcomes, and so presently, it is unlikely that GCRsim or gamma affect anxiety-like and depressive-like behaviors, though additional studies will be needed to confirm this. 

We assessed alterations in startle and PPI responses in relation to GCRsim and gamma irradiation exposure. Pre-pulse inhibition was assessed at multiple white noise tones, but it was at 82 dB that we found significant differences; specifically, F WT mice demonstrated impaired sensorimotor gating in response to both GCRsim (0.75 Gy) and gamma radiation (2 Gy). Similarly, F WT mice receiving 2 Gy gamma also had a reduced startle response, adding evidence of a specific interaction between gamma irradiation, dose, and sex and its effect on the dopaminergic system [25]. While females were solely affected by gamma, males were solely affected by GCRsim, suggesting a combined effect of sex and radiation type. This is not the first time sex differences were found on the startle test, as our previous study on the long-term effects of ^56^Fe found that it reduced startle response in WT males and increased it in females [5]. While we found GCRsim to reduce startle response in WT and Tg males at different tones, we did not find changes in females irradiated with GCRsim or gamma. While future studies are needed to elaborate on five-ion GCRsim’s effect on startle and PPI, a previous study investigated these outcomes in the context of acute proton and gamma irradiation and found that both dose and latency-to-testing affected startle reactivity [26]. Additionally, they found that the acute effects of irradiation were transient; there were no differences between proton and gamma radiation. They also stated that the lack of differences between particles resembled their previous results when comparing ^56^Fe and ^21^Si to controls. Another study involving gamma radiation found a 144% increase in dopamine 3 days after and a 27% reduction in serotonin 30 days later in the hippocampus [27]. This study also concluded that there were transient and slight effects on the central nervous system. Overall, there is much nuance to understanding the effects of radiation type, dose, and time on neurotransmitter pathways. However, we add that our results, though modest, demonstrate biological significance in that a single dose of GCRsim or gamma can modulate dopaminergic functioning in the long term (7 months after exposure). It is important to take these results into consideration with the current literature, given the chronic environment that astronauts will be exposed to.

We observed several radiation-specific effects across locomotor activity, strength and endurance, and motor coordination. On the open field test, there was a significant difference between doses of gamma in M WT mice, where the 2 Gy group traveled less distance. These results partially conflict with another study that found gamma to reduce locomotion compared to shams 6 h post-irradiation; however, we investigated long-term effects, a substantial time difference [28]. Interestingly, in our long-term proton study, there was also a dose difference in M Tg mice, with the 2 Gy group traveling a greater distance than the 0.5 Gy group and with nearly identical distances between sham-irradiated M WT and M Tg mice [2]. Although a direct comparison cannot be made, the closeness in relative biological effectiveness between gamma (1) and protons (1.1) [26], in conjunction with the similarity between the previous and present results, is noteworthy. Though there are distinctions, such as gamma dosage declining to a greater degree with tissue depth [29] and previous studies reporting differences in immune-related parameters [30,31], it may be the case that differential responses are dependent on the parameter tested. While ^56^Fe is a component of GCRsim, in M Tg mice, we found that 0.75 Gy GCRsim increased rearing compared to the 0.5 Gy group, whereas ^56^Fe irradiation caused decreases relative to controls [5]. Though we found no significant differences compared to controls, this opposite directionality further exemplifies the complexity of multiple-ion exposure.

Gamma and GCRsim affected strength and endurance in opposing ways, while gamma irradiation caused a lowering in grip strength in M WT mice, GCRsim caused a significant increase in endurance in M Tg mice. We previously found no differences in either outcome with ^56^Fe; however, 2 Gy of protons did also lower grip strength compared to the 0.5 Gy group in M WT mice. Again, though no direct comparison can be made between the gamma and proton results, the similarity is interesting to note. Protons also reduced endurance in F WT mice, unlike the increase we observed in M Tg mice with GCRsim. Overall, further studies are needed to investigate the effects of GCRsim and gamma on strength and endurance; however, both appear to specifically affect males, but in rather opposite ways.

In this study, the only instance of radiation effects in F Tg mice was on the rotarod test. Since some effects were found in F WT mice, this general “protection” may come from a combination of sex and genotype, though relatively more effects were seen in F Tg mice in the previous ^56^Fe and proton studies but not on the rotarod test. With GCRsim, we found strong evidence of sex dimorphism as radiation impaired motor coordination in F Tg mice but improved it in males. This is partially in line with our ^56^Fe results, as we also found improved coordination in M Tg mice; however, females were not affected [5]. Additionally, these results further differ from protons, where we saw no effects in Tg mice [2]. Gamma had no effect on rotarod results, as also seen in another study assessing delayed effects of a substantially higher dose of gamma irradiation in rats [32].

To our knowledge, this study is the first to investigate Aβ pathology in APP/PS1 mice following GCRsim irradiation. Interestingly, as seen in some of our previous studies using ^56^Fe [5] and protons [2], no radiation-specific changes were seen in fibrillar Aβ burden using Thio S. However, both these previous studies assessed the immunostaining of R1282 in the hippocampus, a polyclonal antibody for Aβ, and found that 0.5 Gy of ^56^Fe induced significantly more pathology in APP/PS1 males compared to the 0.10 Gy dose, whereas both 0.5 Gy and 2 Gy of protons reduced staining in APP/PS1 males compared to sham controls. Given the opposite directionality of staining levels in male APP/PS1 mice between different ion species, it is interesting to note that no differences were seen with monoclonal anti-amyloid antibody 3A1 after GCRsim, which includes both protons and ^56^Fe. On the other hand, insoluble Aβ ELISAs were consistently used across our studies of long-term radiation effects, allowing for direct comparisons. For example, in male APP/PS1 mice, 0.5 Gy ^56^Fe irradiation increased Aβ40 levels; however, no effect was seen with protons [2,5]. On the contrary, 2 Gy proton irradiation reduced Aβ42 levels relative to the 0.5 Gy group, while no significant differences occurred with ^56^Fe [2,5]. Here, we add that, like our previous long-term studies, no differences were found in females; however, unlike those reports, there were also no significant differences in GCRsim males with either test, another example of the complex relationship between ion properties and biological impact. 

In addition to our investigation of changes in AD pathology, we also used brain MRIs to assess whether GCRsim or gamma irradiation induced larger scale changes, and surprisingly, we found many. For example, while sham-irradiated M WT mice had significantly enlarged lateral ventricles in their follow-up scans, those receiving either GCRsim or gamma irradiation did not, suggesting their protective effect on neurodegeneration. Interestingly, the opposite was true in M Tg mice, which demonstrated greater lateral ventricular volumes with GCRsim and no difference in sham controls. Furthermore, F Tg mice were unaffected by GCRsim. While F WT mice were similarly unaffected by gamma, there appeared to be an n-shaped dose–response curve with GCRsim as the 0.5 Gy group displayed an enlargement of ventricles not seen in sham control or the 0.75 Gy group. When assessing changes in the hippocampus, M Tg mice again experienced volumetric changes; interestingly, their post-GCRsim scans revealed increased hippocampal volume, suggesting a possible neurotrophic effect rather than a degenerative one, as seen with the lateral ventricles. However, given that we saw few cognitive effects with GCRsim in M Tg mice and that gamma irradiation was also able to increase hippocampal volume in M WT mice but still impaired spatial memory in the SNYM, another possible explanation is that this enlargement is due to inflammation. For example, there is evidence that radiation causes reactive gliosis (in microglia and astrocytes), and that the hippocampus is radiosensitive [32,33,34]. Additionally, radiation-induced vascular injury causing perivascular edema may be another contributing factor [35]. Investigating the reasoning behind this occurrence may be the subject of future studies. However, apart from 2 Gy gamma, female hippocampi were largely unaffected by radiation in both WT and Tg mice. This was not the case with the cortices of F WT mice. We found a reduction in the cortical volume of F WT mice receiving GCRsim, which, in conjunction with the lateral ventricle enlargement we also observed, clearly points to the degenerative effect of the irradiation that is independent of Aβ pathology. Similarly, gamma irradiation also significantly reduced cortical volume in F WT mice, demonstrating the region’s general vulnerability to radiation. Furthermore, these results may explain some of the results we saw with tests like pre-pulse inhibition. For example, F WT mice receiving 0.75 Gy of GCRsim demonstrated impaired sensorimotor gating that may be due to their reduced cortical volumes. The same relationship was seen in F WT mice receiving 2 Gy of gamma irradiation. Unlike in the hippocampus and lateral ventricles, M Tg mice had no changes in their cortex due to GCRsim. Male WT mice experienced many effects from radiation; however, it is difficult to make clear conclusions as 0.75 Gy of either GCRsim or gamma increased hippocampal volume but decreased cortical volume. Future studies should also investigate this phenomenon, although it may be due to unique region-specific effects of radiation. Additionally, we chose to perform follow-up MRI scans when the mice were 11 months of age, when amyloid pathology and cognitive impairment are well underway, and though we believe this time frame captures enduring effects, future studies may also investigate later time points to assess whether these effects remain.

Although there were nearly no radiation effects seen with heart MRIs, we did find gene expression changes in the heart, indicating that GCRsim and gamma’s impact is more evident at the cellular level. The data reveal a consistent downregulation of *VLCAD* across all irradiated animals and a specific downregulation of *GLUT4* in the high-dose APP/PS1 group. The *GLUT4* gene, responsible for encoding the solute carrier family 2 member 4, is integral for insulin-dependent glucose uptake [36]. While its relevance to myocardial function may be subject to debate, *VLCAD*, which encodes the very-long-chain acyl-CoA dehydrogenase, is necessary for the oxidative metabolism of long fatty acid chains within mitochondria, and thus crucial for optimal myocardial metabolism. Deficiency in *VLCAD*, associated with a genetic metabolic disorder (OMIM 609575), exerts its effects across various organs and systems, with clinical manifestations that include cardiomyopathy and rhabdomyolysis. Although the downregulation of *VLCAD* in the irradiated animals may render them less tolerant to fasting, the actual physiological effects of this observation were not evaluated in this study. Interestingly, while impaired mitochondrial function resulting from *VLCAD* dysfunction could lead to decreased energy production and increased oxidative stress (hallmarks of Alzheimer’s pathology), studies have indicated an upregulation of *VLCAD* (*ACADVL*) in glial cells, correlating with beta-amyloid protein accumulation in Alzheimer’s disease [37]. Further investigation into the effects of GCRsim on *VLCAD* expression in the mouse brain is warranted. However, the enduring alterations in gene expression observed 8 months post-GCRsim exposure suggest the involvement of long-term regulatory mechanisms such as epigenetic modifications. This underscores the necessity for comprehensive exploration into the underlying molecular pathways driving these persistent changes.

Lastly, we found changes in multiple cytokines and a chemokine following radiation exposure. For example, in M Tg mice, compared to sham controls, the plasma levels of IL-2 were higher in 0.75 Gy GCRsim mice, while IL-5 and IL-6 were both lower in the 0.5 Gy group. We also found trends for reductions in IL-10, an anti-inflammatory cytokine, with both doses of GCRsim in M WT mice, and an increase in KC/GRO, a chemoattractant for immune cells, with the highest gamma dose in F WT mice. IL-2 is a cytokine with broad functions but plays essential roles in the immune system, such as driving T-cell growth and increasing natural killer cell cytolytic activity [38]. Taken together, GCRsim appears to have upregulated innate and adaptive immune responses in M Tg mice. IL-6 is another pleiotropic cytokine, which, in addition to provoking acute phase proteins, also mediates chronic inflammation and improves T-cell survival [39]. While it is possible that the decrease in IL-6, and therefore, T-cell survival, may be tied to the elevation of IL-2, which drives its growth, given that these changes were solely in M Tg mice, GCRsim certainly seems to induce a unique immune response. Furthermore, these cytokine ELISAs were similarly used in our previous long-term ^56^Fe and proton studies, allowing us to compare the inflammatory profiles of different types of radiation. For example, while proton irradiation had no effects on IL-2, IL-5, or IL-6 in male APP/PS1 mice, 0.5 Gy ^56^Fe irradiation elevated plasma IL-6 levels; meanwhile, we found a reduction in the case of 0.5 Gy GCRsim. On the other hand, ^56^Fe and proton irradiation were able to induce changes in cytokines that GCRsim and gamma did not [2,5]. 

Overall, we found modest effects of GCRsim and gamma irradiation on behavior, volumes of specific brain regions, and inflammatory cytokines, and more subtle effects on gene expression in the heart. Our data demonstrate that GCRsim and gamma exposure both have the potential to cause long-term impacts on radioresistant tissues, especially in males. For example, M WT mice exhibited radiation-induced cognitive impairments, even in the absence of such effects in their Tg counterparts, similar to the findings in our previous studies [3,5] and likely explained by low levels of estrogen, a neuroprotective hormone. Interestingly, throughout this study, we found several instances where GCRsim irradiation demonstrated a protective effect, such as improved endurance on the wire hang test or increases in hippocampal volume. It is very possible that this form of radiation has the potential to induce neurohormesis [40,41].

However, this study is not without limitations. Firstly, given that this work was carried out in mice, our results are exploratory and point to possible effects that may not perfectly translate to human astronauts. Furthermore, the attrition of F Tg mice, which was likely due to higher levels of amyloid pathology and occurrence of seizures, may have contributed to survivorship bias in those that completed the study. Additionally, chronic exposure to radiation, like that in space, was not a component of this study and may yield different results, especially if combined with simulated microgravity and/or other space travel-related stressors, such as isolation. Future studies would benefit from better capturing these key factors.

## 4. Materials and Methods

### 4.1. Mice

All mice (*n* = 228) were APPswe/PS1dE9 (APP/PS1; stock number 004462) and age-matched wild-type C57BL/6J (WT; stock number 000664) littermate controls. APP/PS1 mice overexpress human APP with the Swedish mutation (K595N/M596L) and mutant Presenilin 1 (PS1-dE9) and develop extracellular beta-amyloid deposits in the brain around 6–7 months of age and microhemorrhages, gliosis, and cognitive deficits by 7–8 months of age. Mice were bred in-house at Brigham and Women’s Hospital and were 2–4 generations from Jackson Laboratory originals. At both BWH and BNL, mice were housed under a 12/12 light/dark cycle with ad libitum access to food (PicoLab Rodent Diet #5053) and water. Mice included in the MRI studies underwent pre-irradiation scans at 3.5 months. At 4 months of age, all mice were shipped to BNL, irradiated after acclimating for a few days, and shipped back to BWH. After returning to BWH, mice underwent quarantine, which included a fenbendazole diet for three weeks and two antibiotic treatments of selamectin (one at the beginning and one at the end of the quarantine period), and aged for 7 months. At 11 months of age, mice either underwent behavioral testing or an MRI scan. At 12 months of age, all mice were euthanized via CO_2_. After a cardiac puncture, mice were perfused with Phosphate-Buffered Saline (PBS). Half of the brain, liver, kidney, spleen, heart, and aorta were fixed in 4% paraformaldehyde (Electron Microscopy Sciences, Hatfield, PA, USA) and half were snap-frozen on dry ice. All animal studies were approved by the Institutional Animal Care and Use Committees at Brigham and Women’s Hospital and Brookhaven National Laboratory. 

### 4.2. Irradiation

A graphic of the experimental timeline can be found in Figure 1. Mice were irradiated in two identical cohorts (*n* = 114 mice per cohort) of male and female, APP/PS1 (*n* = 9 per sex/genotype/dose) and WT (*n* = 6 per sex/genotype/dose) mice, with the first being irradiated in April 2019 and the second in October 2019. Mice were shipped to BNL at 4 months of age and allowed to acclimate for a few days prior to irradiation. For GCRsim irradiation, mice were placed in small plexiglass “mouse hotels”, with one or two other cage-mates. The mice had space to turn around and were provided with enrichment. The hotels were stacked in a grid-like fashion within a frame; hotels with male mice were on the bottom, while hotels with female mice were on the top. Males and females were separated by at least 3 rows of blank hotels. The frame was then carried into the cave for irradiation. GCRsim consisted of 5 different ion species, which were delivered in rapid succession in the following order: protons, Silicon-28, Helium-4, Oxygen-16, Iron-56, and protons again. GCRsim irradiation included 1 GeV/n H+ (0.2 KeV/micron) at an exposure of 0.2625 Gy, 600 MeV/n 28Si (50.4 KeV/micron) at an exposure of 0.0075 Gy, 250 MeV/n 4He (1.6 KeV/micron) at a dose of 0.135 Gy, 350 MeV/n 16O (20.9 KeV/micron) at an exposure of 0.045 Gy, 600 MeV/n 56Fe (173.8 KeV/micron) at an exposure of 0.0075 Gy, and 250 MeV/n H+ (0.4 KeV/micron) at an exposure of 0.2925 Gy, totaling 0.75 Gy. With a Poisson distribution, this equates to an estimated average of 735 million, 93,200, 53.5 million, 1.36 million, 27,500, and 462 million particle traversals per square cm, respectively. Following irradiation, mice were returned to their home cages. The sham group was loaded into the hotels and the hotels were stacked in the frame, but the frame was not carried into the cave. Mice returned to BWH within 2–3 days following irradiation. 

For gamma irradiation, mice were loaded into pie-shaped plexiglass holders, which fit 8 mice, 1 in each compartment. The tubes were placed into a foam holder and placed into the irradiator. Males and females were irradiated separately. The irradiation with gamma rays from a Cesium-137 source consisted of a photon energy of 662 KeV (0.8 keV/micron) at 2 Gy at a rate of 112 mGy/minute. With a Poisson distribution, this equates to an estimated average of 1.56 billion traversals per square cm.

### 4.3. Behavioral Tests 

Between both cohorts, 124 mice completed various behavioral tests at 11 months of age. Cohort 1 mice were completed testing in December of 2019 while cohort 2 mice completed testing in July of 2020. For all tests, females and males were tested separately. 

#### 4.3.1. Open Field (OF)

The OF test measures locomotor activity and anxiety-like behavior. Mice were placed in 27 × 27 cm test chambers (Med Associates, St. Albans, VT, USA) and allowed to explore for one hour. A computer-assisted infrared tracking system computed the total distance traveled, time and distance spent in the center, and rearing. Total distance traveled provides a measure of locomotor activity, while time and distance spent in the center is a measure of anxiety-like behavior. 

#### 4.3.2. Grip Strength (GS)

The GS test is a measure of forepaw grip strength. After acclimating to the room for 15 min, mice were picked up and allowed to grab the grip strength meter with their forepaws. Once both paws were on the meter, mice were slowly pulled horizontally until they let go, and the force was recorded. Mice were given 3 consecutive trials with a 10 s inter-trial interval in a holding cage. 

#### 4.3.3. Wire Hang (WH)

The WH test is a measure of endurance. The apparatus consisted of a horizontal wire placed 50 cm on top of a cage with 4 cm of bedding. The wire was secured to support posts with tape. Laminated paper disks were used to ensure that mice could not climb down the support posts. After acclimating to the room for 15 min, mice were given 3 consecutive trials (maximum 3 min per trial) with 30 s inter-trial intervals in a holding cage. Mice were placed on the top of the wire, and once they gripped the wire with all 4 paws, they were gently flipped so that they were hanging upside down. The latency to fall was recorded. 

#### 4.3.4. Rotarod

A rotarod measures motor coordination and motor learning and consists of a training phase followed by 2 test phases. During the training phase, mice acclimated to the rotarod for 5 min at 4 rotations per minute (rpm), and the number of falls was recorded. The test phases occurred immediately after all mice had finished the training phase. In the test phases, the speed of the rotarod increased from 4 to 40 rpm over 3 min and stayed at 40 rpm for an additional 2 min. The latency to fall was recorded. 

#### 4.3.5. Elevated Plus Maze (EPM)

The EPM consists of a plus-shaped maze with 2 open arms and 2 closed arms. The willingness (or lack thereof) of the mouse to explore open arms is a measure of anxiety-like behavior. Each arm was 30 cm long and 5 cm wide, and the apparatus was 50 cm above the floor. The test was conducted in dim ambient lighting. Mice were placed in the center of the 4 arms and allowed to explore for 5 min. The time and distance traveled in the open arms were recorded via Ethovision (https://www.noldus.com/ethovision-xt, 7 July 2024). 

#### 4.3.6. Spatial Novelty Y Maze (SNYM)

The SNYM consists of a Y-shaped maze with a start arm, a left arm, and a right arm. It is a measure of spatial memory. During the training phase, either the left or right arm was blocked. The specific arm blocked was kept balanced in each sex/genotype/dose group. Mice explored the two arms for 3 min, and then removed from the maze for 2 min. For the test phase, all 3 arms were unblocked and cleaned, and the mice were allowed to explore all 3 arms for a final 3 min. A cognitively intact mouse should spend more time exploring the novel arm in the test phase. The time and distance the mouse traveled in each arm were recorded by Ethovision. 

#### 4.3.7. Tail Suspension Test (TST)

The TST provides a measure of depressive-like behaviors. A piece of clear scotch tape was folded over the end of the mouse’s tail. The mice were hung by the tape on their tails for 6 min, and the time immobile was recorded using Ethovision. Time spent immobile is a measure of stress responsivity. 

#### 4.3.8. Startle Reactivity

Mice were placed in restrictive holders (Med Associates, St. Albans, VT, USA) next to speakers and on top of transducer platforms, which measure the mouse’s response to tones. The speakers, holders, and restrainers were in 25″ × 16.5″ × 15.5″ startle chambers that were lined with 0.75″ of foam. After a 5 min acclimation to the restrainers, mice completed 10 blocks of 11 trials each, for a total of 110 trials. Within each block, white noise acoustic stimuli were presented for 40 ms each in a random order. The stimuli were 20, 30, 40, 50, 60, 70, 80, 90, 100, 110, or 120 dB white noise tones. The inter-trial interval ranged from 10 to 20 s, with a mean of 15 s. The mouse’s startle response was recorded each millisecond for 150 ms after each stimulus. 

#### 4.3.9. Pre-Pulse Inhibition

Pre-pulse inhibition (PPI) is a measure of sensorimotor gating. Mice were placed into the same restrainers and chambers (Med Associates) used for the startle reactivity test. After a 5 min acclimation period, mice underwent 10 blocks of 6 different types of trials, which were either null (no stimuli), startle (120 dB), startle with a pre-pulse of either a 65 dB, 75 dB, or 85 dB 5000 Hz tone, or an 85 dB pre-pulse with no startle tone. Each trial began with a 50 ms null period, where baseline movements were recorded. In trials with the pre-pulse and the startle tone, the pre-pulse lasted for 20 ms; after 100 ms, the startle tone was presented for 40 ms. Responses were recorded every ms until 140 ms from the onset of the startle tone. The inter-trial interval ranged from 10 to 20 s, with an average of 15 s. The percent inhibition was calculated as follows: (100 − pre-pulses + startle)/(startle) × 100 and is a measure of sensorimotor gating. 

##### Contextual Fear Conditioning (CFC)

The CFC test measures fear memory. The testing apparatus consisted of operant chambers with plexiglass side walls. The floors consisted of steel bars spaced 1 cm apart, capable of delivering shocks. During the training phase, mice acclimated to the chambers for 2 min and then received 22 s 0.5 mA shocks (Med PC) with a 2 min inter-trial interval. Chambers were cleaned with 70% ethanol in between mice. After 24 h, mice were placed back into the chambers for the context test. They were allowed to explore for 5 min but did not receive a shock. Once all mice had completed the context test, the chambers were modified for the alternate context test; white, opaque inserts were placed into the walls and floor of the chambers such that the walls were curved, altering the interior appearance. Mice were placed in the chambers and allowed to explore for 5 min but did not receive shocks. The chambers were cleaned with Windex in between mice to provide a different ambient odor. Media Recorder was used to record videos of all 3 phases, and freezing behavior was scored using Ethovision.

### 4.4. MRI Imaging and Analysis

Mice underwent MRI scans before irradiation and 7 months post-irradiation. 3DSlicer (slicer.org) [42] was used to import and analyze brain and heart scans from a 3T Bruker machine. T1 and T2 axial scans of the brain were overlayed, and the segmentation tool was used to obtain volumetric data for the hippocampus, cortex, and lateral ventricles. Short-axis FLASH MRIs were used for heart imaging. The images corresponding to left ventricular end-diastole (LVED) and end-systole were used for segmentation, and the resulting difference in volume represented the stroke volume. The stroke volume divided by the LVED volume, multiplied by 100, gave the ejection fraction. The ruler tool was used on the LVED image to measure the posterior, anterior, septal, and lateral walls, and the average of the measurements determined the left ventricular wall thickness in end diastole. To calculate the mass of the left ventricle, the left ventricular wall thickness was multiplied by 1.05 g/cm^3^, the conversion coefficient. During imaging, mice were anesthetized using vaporized isoflurane, and their eyes were lubricated with gel to prevent dryness. Additionally, a respiration sensor was placed to monitor mice’s breathing rate throughout the scan and to inform adjustments to the anesthesia. Following completion of the scan, mice recovered on a heating pad until they could ambulate spontaneously.

### 4.5. MSD Aβ Protein ELISA

Frozen hemibrains were homogenized with ice cold Tissue Protein Extraction Reagent (TPER) buffer (Fisher Scientific, Hampton, NH, USA) with Protease Inhibitor Cocktail Set I and Phosphatase Inhibitor Cocktail Set III (Calbiochem #524627 and #539131 respectively), sonicated, and spun at 41,000 rpm for 1 h at 4 degrees Celsius. The supernatant was collected as the soluble Aβ fraction, and the pellet was resuspended, homogenized, and incubated for 3 h in 5 M Guanidine HCl at pH 8. The Guanidine HCl had the same protease and phosphatase inhibitors as the TPER and at the same concentrations. The samples were then spun at 41,000 rpm for 1 h, and the supernatant was collected as the insoluble Aβ fraction, which was diluted to 1:10,000. 

Insoluble Aβ40 and 42 was quantified using the V-PLEX AB Peptide Panel 1 (4G8) kit (Meso Scale Diagnostics #K15199E-1, Rockville, MD, USA). The assay was performed per the manufacturer’s instructions. 

### 4.6. MSD Cytokine ELISA

The Pro-inflammatory Panel 1 (mouse) V-PLEX ELISA kit (Meso Scale Diagnostics # K15048D, Rockville, MD, USA) was used to quantify plasma cytokine levels of IFN-γ, IL-2, IL-5, IL-6, IL-10, KC/GRO and TNF-α. Plasma samples were diluted 1:2 with diluent provided by the kit, and ELISAs were conducted following the procedures given by the kit insert.

### 4.7. Immunohistochemistry, Histology, and Quantification

Mouse brains were embedded in agarose before being sectioned with a vibrating microtome and placed into 12-well plates containing 0.082% sodium azide (Sigma-Aldrich, St. Louis, MO, USA) solution. Immunolabelling was carried out using the ABC ELITE method (Vector Laboratories, Burlingame, CA, USA). Monoclonal antibody 3A1 [1:200, gift from Dr. Brian O’Nuallain] was used to assess general Aβ staining. One percent aqueous Thioflavin S (Thioflavin S; Sigma-Aldrich, St. Louis, MO, USA) was used to visualize fibrillar amyloid in plaques and blood vessels. Immunoreactivity of 3A1 and Thioflavin S in the hippocampus (HC) was imaged using a Zeiss Axiovert Imager.A1 with an AxioCam MRc5 camera and quantified using ImageJ. An average of 3 consecutive, equidistant sections were used for both the analyses. Heart and kidney tissues were fixed in 4% paraformaldehyde, embedded in paraffin, and sectioned at 5 μm. Sections were deparaffinized in xylene and rehydrated through graded alcohols to water. For the detection of collagen fibers, sections were stained with Hematoxylin and Eosin (H&E) and Picrosirius Red (PSR) to assess general tissue architecture and collagen deposition, respectively. For H&E staining, tissue sections were first deparaffinized and rehydrated. Hematoxylin was applied for 5–10 min to stain the nuclei, followed by rinsing and differentiation in acid alcohol. Eosin was then applied for 1–3 min to stain the cytoplasm. The sections were dehydrated and mounted for examination. For PSR staining, tissue sections were similarly deparaffinized and rehydrated, then stained with Picrosirius Red solution for 1 h. The slides were rinsed in acidified water, dehydrated, and mounted to assess collagen content. Additionally, slides of both heart and kidney tissues were stained with Masson’s Trichrome to further evaluate fibrotic changes. Stained sections were examined under light microscopy. Representative photographs were taken using an Olympus U-MDOB3 microscope with 2× and 10× objectives for heart sections and 1.25× and 10× for kidney sections, using H&E and PSR stains to visualize and quantify fibrosis. The extent of fibrosis was quantified using a computerized analysis with Image Pro Plus version 7.0 software. This involved calculating the mean percentage of fibrosis from four representative images at 10× magnification for each tissue section and averaging these values to estimate the overall tissue fibrosis.

### 4.8. Heart and Kidney Gene Expression

For gene expression analysis, total RNA from heart and kidney tissue was isolated using the miRNeasy mini kit (Qiagen #217004, Hilden, Germany), followed by cDNA synthesis with the High-Capacity cDNA Reverse Transcription Kit (Thermo Fisher Scientific #4368814, Waltham, MA, USA). Real-time PCR assays were performed on a QuantStudio 6 system (Thermo Fisher) using SYBR Select Master Mix chemistry (Thermo Fisher #4472918). Comparative gene expression analysis was conducted using the Delta-Delta Ct Method, with GAPDH serving as the reference gene. All reactions were performed in triplicates. Statistical significance was determined using the *t*-test with false discovery rate (FDR) correction for multiple testing.

### 4.9. Statistical Analyses

All of the data are expressed as the mean ± SEM. A value of *p* < 0.05 was considered significant, and *p* < 0.1 was considered a notable trend for all statistical tests. Data were analyzed as reported in the specific results and figure captions. Generally, 2-way and 1-way ANOVAs were used to examine the effects of sex, dose, and genotype on each endpoint. Following ANOVAs, a family of pairwise comparisons was made, and the family wise error rate was controlled with Tukey’s correction. Sham-irradiated control groups (ex: M Tg vs. F Tg) were also compared independent of ANOVAs using two-tailed, unpaired *t*-tests or Mann–Whitney U tests if the data were not normally distributed. Additionally, WT mice were shared between GCRsim and gamma analyses. StatView version 5.0 (SAS Institute, Cary, NC, USA) and Prism version 9.3.1 (GraphPad, San Diego, CA, USA) software were used for analysis.

## Figures and Tables

**Figure 1 ijms-25-08948-f001:**
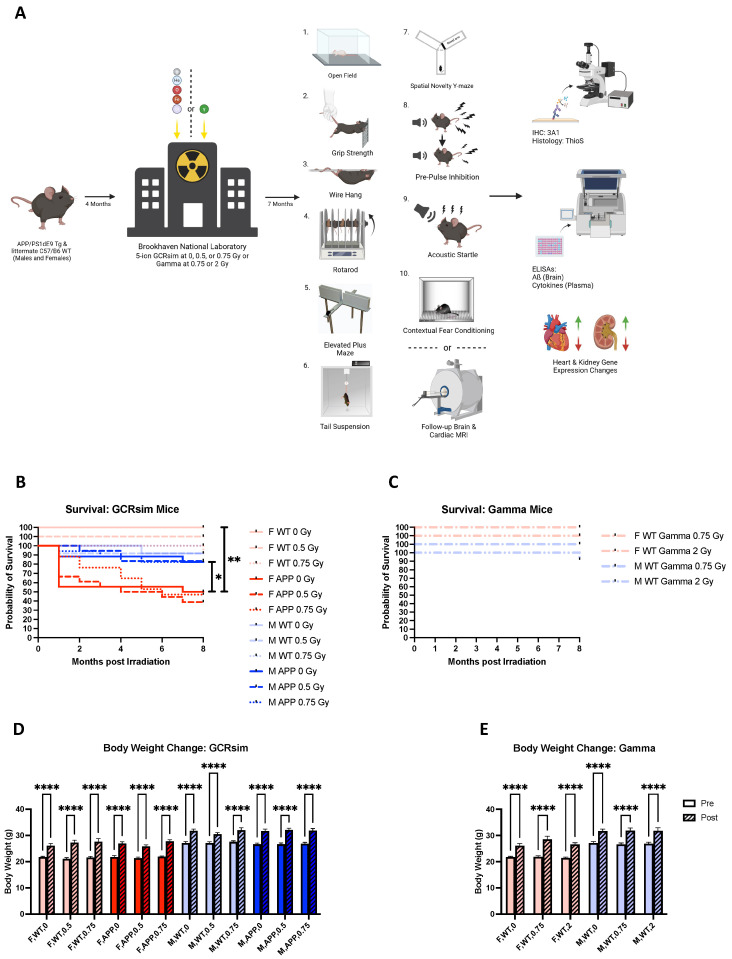
(**A**) Study timeline: At 4 months of age APPswe/PS1dE9 Tg and WT mice were transported to and from Brookhaven National Laboratory for GCRsim (0, 0.5, or 0.75 Gy) or gamma (0, 0.75, or 2 Gy) irradiation (*n* = 7–15 mice/group). A select group of mice underwent pre-irradiation brain and cardiac MRI scans at 3.5 months of age and 7 months post-irradiation. These mice received follow-up scans. Mice not involved in MRI studies participated in behavioral testing instead. At 12 months of age, mice were euthanized, and tissues were harvested for immunohistochemistry, histology, gene expression analyses, and ELISAs. Created with BioRender.com. (**B**) Survival curve of mice included in GCRsim analyses. (**C**) Survival curve of mice receiving gamma irradiation. Sham-irradiated controls are included in (**A**). Survival curve analyses in (**B**,**C**) utilized the log-rank test. (**D**,**E**) Differences in body weight in GCRsim (**D**) and gamma (**E**) mice at 12 months compared to 4 months of age. Body weight analyses were performed using repeated measures 2-way ANOVAs with Šídák’s multiple comparisons corrections. *: *p* < 0.05, **: *p* < 0.01, ****: *p* < 0.0001.

**Figure 2 ijms-25-08948-f002:**
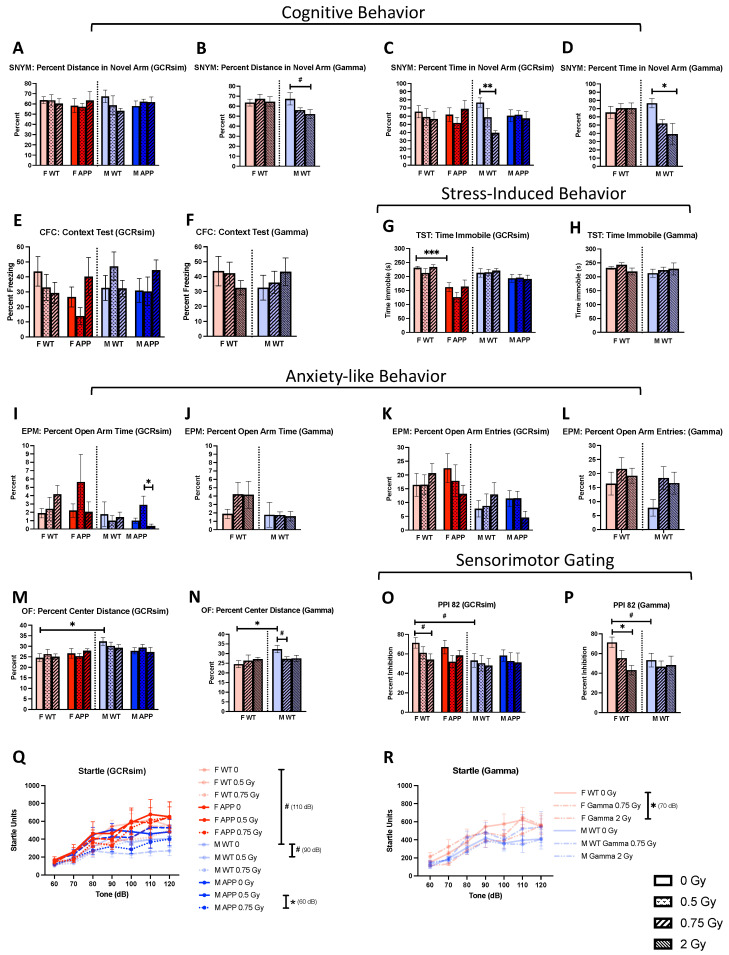
Differences in outcomes from behavioral tests (*n* = 4–11 mice/group). (**A**–**D**) Differences in the percentage of total distance (**A**,**B**) and time (**C**,**D**) spent in the novel arm of the spatial novelty Y maze (SNYM). (**E**,**F**) No significant differences were found in the context test. (**G**,**H**) Depressive-like behavioral changes were assessed using the tail suspension test. (**I**–**L**) Differences in the percentage of time (**I**,**J**) spent in the open arm of the elevated plus maze (EPM) as well as the number of entries (**K**,**L**). (**M**,**N**) Radiation and baseline sex effects on the distance traveled in the center of the open field. (**O**,**P**) The pre-pulse inhibition test (PPI) was used to assess sensorimotor gating and revealed several differences at a pre-pulse of 82 decibels (dB). (**Q**,**R**) Startle reactivity was compared at multiple white noise tones and differences found at each tone are indicated. Data were analyzed via 2-way ANOVAs followed up with 1-way ANOVAs with Tukey comparisons and planned, unpaired, 2-tailed *t*-tests between sham groups as necessary. Non-parametric data were analyzed using the Kruskal–Wallis test with Dunn’s multiple comparisons correction and/or Mann–Whitney U tests. Dotted lines indicated males and females were analyzed separately. #: *p* < 0.1, *: *p* < 0.05, **: *p* < 0.01, ***: *p* < 0.001.

**Figure 3 ijms-25-08948-f003:**
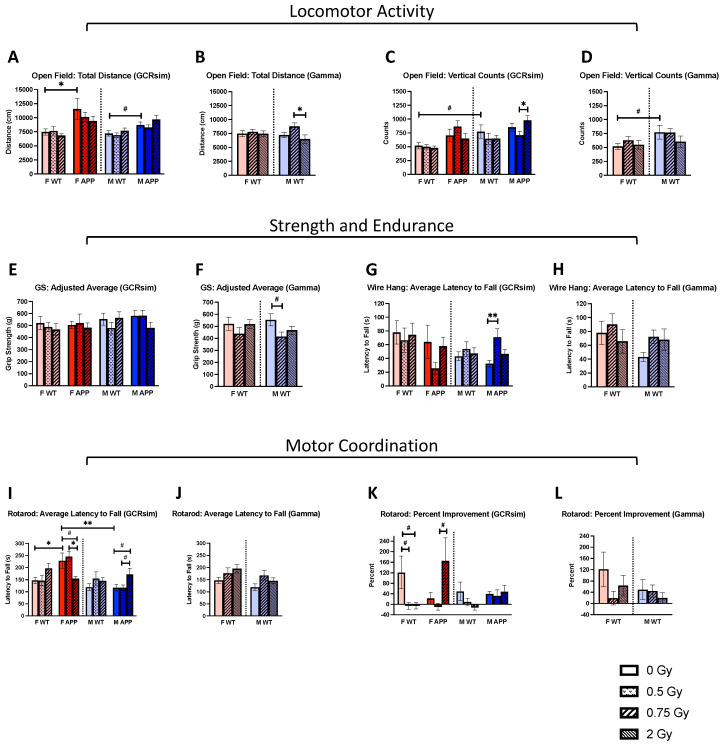
Differences in outcomes from non-cognitive tests (*n* = 4–11 mice/group). (**A**–**D**) The total distance traveled and vertical counts during the open field test were compared across groups. (**E**–**H**) Strength and endurance were assessed using the grip strength and wire hang test, respectively. (**I**–**L**) The latency to fall (**I**,**J**) and percent improvement (**K**,**L**) on the rotarod test determined differences in motor coordination. Data were analyzed via 2-way ANOVAs followed up with 1-way ANOVAs with Tukey comparisons and planned, unpaired, 2-tailed *t*-tests between sham groups as necessary. Non-parametric data were analyzed using the Kruskal–Wallis test with Dunn’s multiple comparisons correction and/or Mann–Whitney U tests. Dotted lines indicate that males and females were analyzed separately. #: *p* < 0.1, *: *p* < 0.05, **: *p* < 0.01.

**Figure 4 ijms-25-08948-f004:**
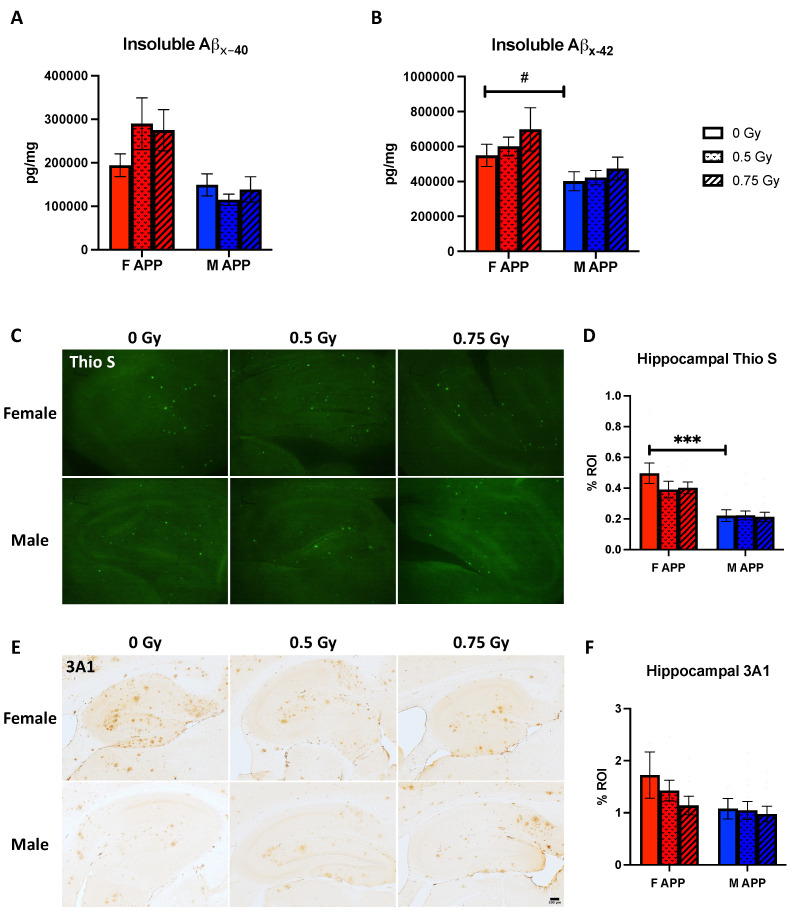
(**A**,**B**) Insoluble Aβx-40 (**A**) and Aβx-42 (**B**) levels were quantified using an MSD 4G8 Aβ-triplex ELISA. (**C**,**D**) Thio S staining (**C**) and quantification (**D**) of Aβ in the hippocampus. (**E**,**F**) 3A1 immunohistochemical staining (**E**) and quantification (**F**) of hippocampal beta-amyloid. Data were analyzed via 2-way ANOVAs followed up by 1-way ANOVAs with Tukey comparisons and planned (*n* = 7–15 mice/group), 2-tailed *t*-tests between sham groups as necessary. Non-parametric data were analyzed using the Kruskal–Wallis test with Dunn’s multiple comparisons correction and/or Mann–Whitney U tests. #: *p* < 0.1, ***: *p* < 0.001.

**Figure 5 ijms-25-08948-f005:**
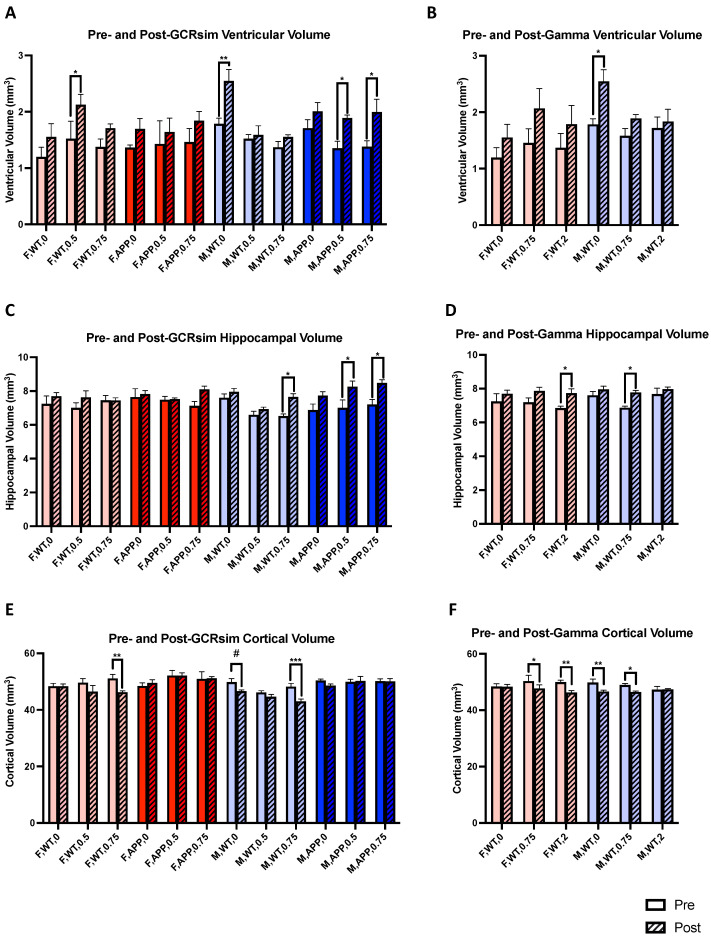
(**A**,**B**) Analyses of paired pre- and post-irradiation lateral ventricle volumes in GCRsim (**A**) and gamma (**B**) irradiated mice using T1 and T2 brain MRI scans (*n* = 3–4 mice/group). (**C**–**F**) Analyses similar to (**A**,**B**), however, with hippocampal (**C**,**D**) and cortex (**E**,**F**) volumes. Repeated measures 2-way ANOVAs with Šídák’s multiple comparisons corrections were used for analyses. #: *p* < 0.1, *: *p* < 0.05, **: *p* < 0.01, ***: *p* < 0.001.

**Figure 6 ijms-25-08948-f006:**
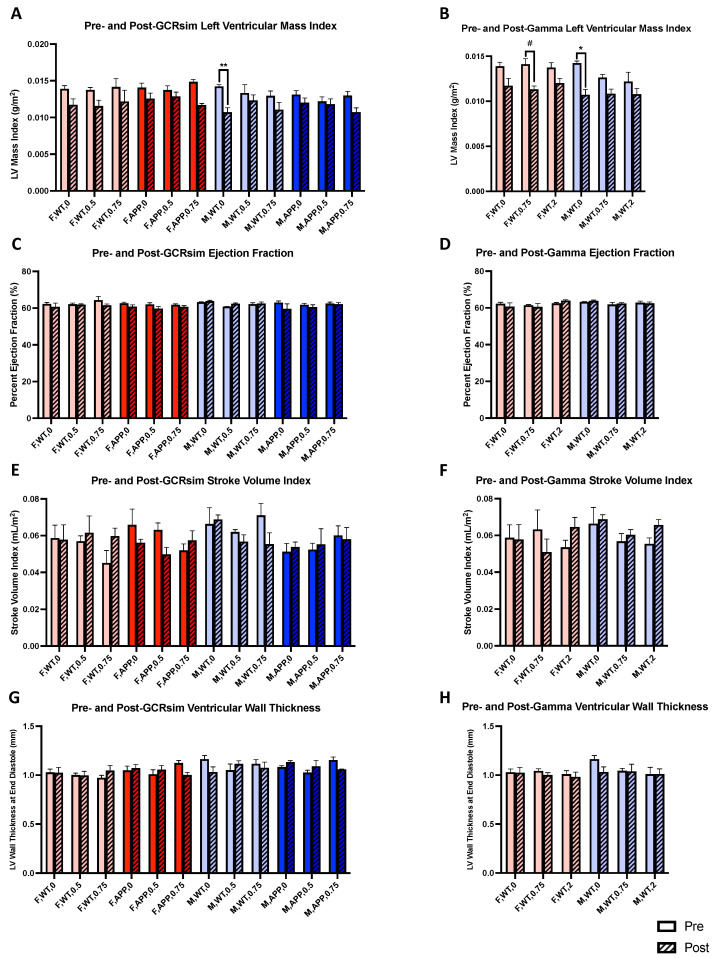
MRI analyses of the heart (*n* = 3–4 mice/group). (**A**,**B**) Analyses of paired pre- and post-irradiation left ventricular mass index in GCRsim (**A**) and gamma (**B**) irradiated mice using short-axis FLASH MRI scans. (**C**–**H**) Analyses similar to (**A**,**B**), however, with ejection fraction (**C**,**D**), stroke volume index (**E**,**F**), and left ventricular wall thickness (**G**,**H**). Left ventricular wall thickness was calculated by averaging the measurements of the posterior, anterior, septal, and lateral walls. Left ventricular mass was calculated by multiplying the left ventricular wall thickness by 1.05 g/cm^3^, the relative density of myocardium and the conversion coefficient. The difference in volumes between left ventricular systole and diastole was used for stroke volume, which, when divided by the end diastolic volume and multiplied by 100, gave the ejection fraction. Repeated measures 2-way ANOVAs with Šídák’s multiple comparisons corrections were used for analyses. #: *p* < 0.1, *: *p* < 0.05, **: *p* < 0.01.

**Figure 7 ijms-25-08948-f007:**
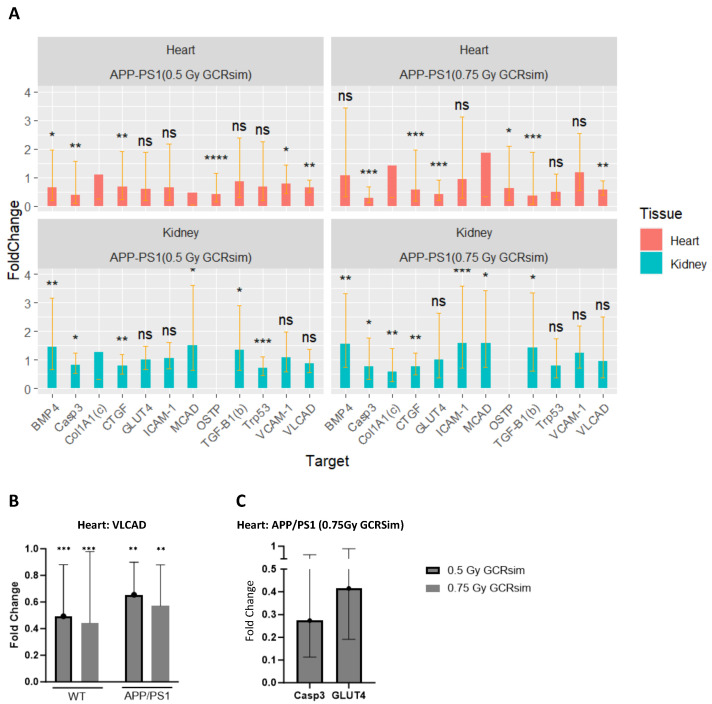
(**A**) Overview of the differentially expressed genes in heart and kidney on GCRsim-treated mice. All bars are referenced to APP/PS1 sham controls. Values between 0 and 1 denote negative fold changes. (**B**) Expression of *VLCAD* gene in the heart tissue of GCRsim-treated WT and APP/PS1 animals, compared to sham-irradiated controls. (**C**) Relative expression of *Casp3* and *GLUT4* in heart tissue of high-dose GCRsim-treated APP/PS1 animals, compared to sham controls. ns: no significance, *: *p* < 0.05, **: *p* < 0.01, ***: *p* < 0.001, ****: *p* < 0.0001.

**Figure 8 ijms-25-08948-f008:**
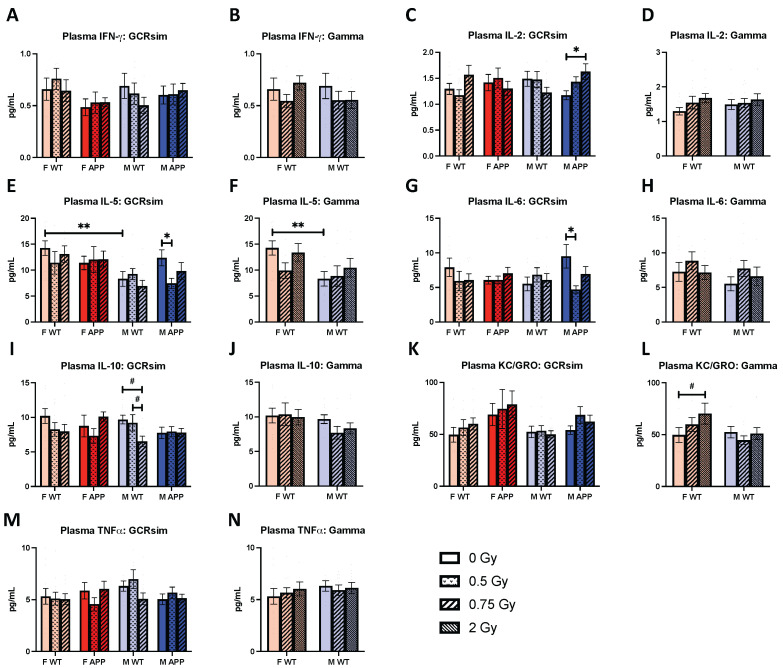
Plasma levels of 7 inflammation-related cytokines measured via MSD ELISA, (**A**,**B**) IFN-γ, (**C**,**D**) IL-2, (**E**,**F**) IL-5, (**G**,**H**) IL-6, (**I**,**J**) IL-10, (**K**,**L**) KC-GRO, and (**M**,**N**) TNF-α (*n* = 7–15 mice/group). GCRsim affected levels of IL-2, IL-5, IL-6, and IL-10 in males. KC/GRO was both the only cytokine altered in females and the only one impacted by gamma irradiation. Data were analyzed via 2-way ANOVAs followed up with 1-way ANOVAs with Tukey comparisons and planned, unpaired, 2-tailed *t*-tests between sham groups as necessary. Non-parametric data were analyzed using the Kruskal–Wallis test with Dunn’s multiple comparisons correction and/or Mann–Whitney U tests. #: *p* < 0.1, *: *p* < 0.05, **: *p* < 0.01.

## Data Availability

All data from this study will be made available through the Ames Space Life Sciences Data Archive and by reasonable request to the authors.

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
