# Peer review of "Long-Term, Sex-Specific Effects of GCRsim and Gamma Irradiation on the Brains, Hearts, and Kidneys of Mice with Alzheimer’s Disease Mutations"

_ijms, 2024, doi:10.3390/ijms25168948_

Round 1

Reviewer 1 Report

Comments and Suggestions for Authors

This work targets the Radiation effects of Cosmic radiation in space in vivo. 

The authors perform elaborate experiments using the facilities of Brookhaven National Laboratories at Upton NY. The radiobiological effects on mice in most cases clear especially for standard gamma rays exposure.

The main problem is to explain the basis of these effects for GCRism as the authors in a rather awkward way choose to use. The irradiation with different high LET ions is not clear. How rapid succession was? Details must be provided protons, Silicon-28, Helium-4, Oxygen-16, Iron-56, and protons again but why?? 

How is it possible to explain effects when you combine high LET Radiations? 

The authors should try to show some initial events of damage induction like for Feor other high LET particles actually done in the same place at BNL. For example see :

 1. https://pubmed.ncbi.nlm.nih.gov/17723001/

2. https://pubmed.ncbi.nlm.nih.gov/36682835/

And others. 

See also some other specific suggestions :

Abstract: Generally, it covers the major aspects of the study in a well-clarified way, but:  

Please note that the abstract says that the mice were evaluated at 11 months, whereas in the introduction and results, they mention “at 11-12 months” or “12 months” (after 7-8 months) – Ensure consistency. 

Introduction: 

1. Line 52: The brain and heart primarily consist of cells that do not readily divide, such as neurons and cardiomyocytes, respectively.

2. Line 56: investigated, not dissected. 

3. To ensure a better transition between the ideas in the introduction, authors could write a small transition paragraph for AD and how radiations could be a risk for developing AD. 

4. Line 80: Authors could mention 1 sentence about the APPswe/PS1dE9 mice that are a model for studying AD. 

5. Authors could mention more about the biological effects of the radiation types they mention. 

Figure 1: 

1. Authors could mention the date they used BioRender.com

2. Figure should also have a key legend, where the abbreviations are explained, even though they were explained in the text. 

Results: 

2.1 is ok

Unit 2.2 comments: 

1. Authors could provide a sentence describing what is tested by the behavioral tests used – details are provided in the materials and methods section, but one small sentence could be mentioned in the results too, because materials and methods are mentioned later in the paper.    

2. Authors should explain the rationale for choosing specific doses for GCRsim and gamma irradiation. Are these doses representative of potential human exposure in space?

Unit 2.3 Comments 

1. Once again, a brief sentence for the test 

2. line 199 in figure 3: were, not was. 

Unit 2.4 Comments

1. Explain the abbreviation: Aβ = amyloid beta protein

2. line 211: introduce a coma after the word mice. 

Unit 2.5 Comments

1. Authors could include representative MRI images in this section or as supplementary material.  

2. Lines 247 and 248: Authors must write the sentences separately, to improve grammar. 

3. Line 264: gamma radiation

Unit 2.6 Comments

1. Line 273: gamma radiation, not plain gamma. 

2. Line 291 in figure 6, the authors could explain more the coefficient thjey mention. 

Unit 2.7 Comments 

1. Why ICAM1, Casp3, GLUT4, MCAD, VLCAD, IL17, and Trp53?

2. Line 327: gene expression regulation

3. When line 328 begins, there is a gap there. It’s like jumping to another idea. Authors should link the fibrosis staining they performed to the COL1A1 gene they mentioned before – this gene is a marker of fibrosis, and they should clearly mention that, with a reference. They should also mention what marker did they use to stain for fibrosis.  

Unit 2.8 Comments

1. Line 336: gamma radiation or gamma irradiation, not plain gamma

2. Line 337: a coma after the world cytokines. 

3. Line 341: mention the result for the females in a separate sentence. 

Discussion:

1. "Overall, our behavioral analysis results from both GCRsim and gamma irradiation provide further evidence of sex differences in response to radiation, with males being disproportionately affected across our battery of tests." - this could replace the first sentence in lines 382-384, to improve clarity. 

2. Authors should discuss the implication of lower survival rates in sham-irradiated Tg females compared to their WT and male counterparts. Why does this difference exist? 

3. Authors should also discuss the finding that WT mice, especially males, exhibited more radiation-induced cognitive impairments. Explore possible reasons why Tg mice did not show similar effects. 

4. Regarding the MRIs section in the discussion, starting in line 490, maybe the authors could consider discussing the sensitivity of MRI in detecting subtle changes and whether longer follow-up periods might yield different results, providing relevant references.

5. Discuss on the radiation doses used. 

6. Discuss on the follow-up period the authors chose to study the mice after irradiation. 

Materials and Methods:

1. Line 743: the correct word is Celsius. 

2. Line 744: temperature of incubation?

3. Line 762: more details on the antibody and mention the composition of the buffer they used to dilute the antibody. 

4. Line 771: More details on H-E staining, e.g. a brief mention of the steps and of duration of staining.

5. Lines 775 -776: The magnifications mentioned are most probably of the objectives. The authors should clarify that. For the total magnification, they should multiply with the magnification power of the ocular lenses (eyepieces) of their microscope. Also they should mention the type of light microscope. 

General: I think that the authors should increase the reference count to at least 50. Also, they should include recent similar studies like “ Complex 33-beam simulated galactic cosmic radiation exposure impacts cognitive function and prefrontal cortex neurotransmitter networks in male mice”. 

In acknowledgments, some journals just want the names and not titles like Dr. etc

Comments on the Quality of English Language

The English need minor revisions. 

Reviewer 2 Report

Comments and Suggestions for Authors

In this article, authors aimed to explore the potential sex-specific interactions between galactic cosmic ray simulation (GCRsim), radioresistant (postmitotic) tissues and Alzheimer’s Disease (AD) pathology. They performed several behavioural tests together with biochemical analysis. Experimental design is excellent, and the topic is interesting since long-term effects on cognition using this novel GCRsim has not been tested before. However, the single-dose irradiation limits the extrapolation of long-term space travel.

Results are quite hard to comprehend since not always display a dose-dependent response and in some cases AD are even more resistant to GCRsim. Moreover, the sexual dimorphism is also present which further increase the complexity of the interaction.

There are some missing data in several figures:

- There is no information in Figure 1C, 2R.

-  Figure 1B Y-axis has 3x 100 value.

- Figure 1, 2, 3 lack significant symbols in some plots.

- Font size is not always homogeneous (like in Figure 3).

- In some cases, a low dose of GCRsim is particularly protective (compared with no-irradiated or higher dose): which explanation do authors have? Can hormesis take place and be a potential treatment to boost anti-GCRsim response?

- Author should comment results from 3A-3I in the manuscript before 3J in Results.

- In the line 208, authors stated that is surprising the absence of differences of radiation of AB solubility. Is there any article about GCRsim and AD?

- In the lines 209-210, authors stated that there are sex-differences in insoluble AB42 levels but the p-value is 0.09 (not significant). Another term would describe better this comparison (a trend, for example).

- In Figure S1I in WT group there are 2 potential outliers: Did authors perform an outlier test?

- Do authors have a hypothesis of M APP resistance to GCRism in Figure 2C?

Round 2

Reviewer 1 Report

Comments and Suggestions for Authors

The revised version of the Manuscript stands now much better. The authors have carefully addressed all problems. 

Comments on the Quality of English Language

The revision of this work has improved the Manuscript significantly. 

Reviewer 2 Report

Comments and Suggestions for Authors

Authors have made the proper corrections in the reviewed manuscript.